# Synthesis and Characterization of Metallopolymer Networks Featuring Triple Shape-Memory Ability Based on Different Reversible Metal Complexes

**DOI:** 10.3390/polym14091833

**Published:** 2022-04-29

**Authors:** Josefine Meurer, Thomas Bätz, Julian Hniopek, Milena Jäger, Stefan Zechel, Michael Schmitt, Jürgen Popp, Martin D. Hager, Ulrich S. Schubert

**Affiliations:** 1Laboratory of Organic and Macromolecular Chemistry (IOMC), Friedrich Schiller University Jena, Humboldstr. 10, 07743 Jena, Germany; josefine.meurer@uni-jena.de (J.M.); thomas.baetz@uni-jena.de (T.B.); milena.jaeger@uni-jena.de (M.J.); stefan.zechel@uni-jena.de (S.Z.); martin.hager@uni-jena.de (M.D.H.); 2Jena Center of Soft Matter (JCSM), Friedrich Schiller University Jena, Philosophenweg 7, 07743 Jena, Germany; 3Institute of Physical Chemistry (IPC), Friedrich Schiller University Jena, Helmholtzweg 4, 07743 Jena, Germany; julian.hniopek@uni-jena.de (J.H.); m.schmitt@uni-jena.de (M.S.); juergen.popp@uni-jena.de (J.P.); 4Abbe Center of Photonics, Friedrich Schiller University Jena, Albert-Einstein-Straße 6, 07745 Jena, Germany; 5Leibniz Institute of Photonic Technology, e. V. Jena, Albert-Einstein-Str. 9, 07745 Jena, Germany

**Keywords:** smart materials, responsive polymers, triple shape-memory, metallopolymers, metal ligand interaction

## Abstract

This study presents the synthesis and characterization of metallopolymer networks with a triple shape-memory ability. A covalently crosslinked polymer network featuring two different additional ligands in its side chains is synthesized via free radical polymerization (FRP). The subsequent addition of different metal salts leads to the selective formation of complexes with two different association constants (*K_a_*), proven via isothermal titration calorimetry (ITC). Those two supramolecular crosslinks feature different activation temperatures and can act as two individual switching units enabling the fixation and recovery of two temporary shapes. The presented samples were investigated in a detailed fashion via differential scanning calorimetry (DSC), thermogravimetric analysis (TGA), and FT-Raman spectroscopy. Furthermore, thermo-mechanical analyses (TMA) revealed excellent dual and triple shape-memory abilities of the presented metallopolymer networks.

## 1. Introduction

Recently, intelligent materials have become more and more important. The use of stimulus responsive polymers can bring several advantages and improvements in many areas [1,2,3]. By combining metal complexes with polymers, it is possible to create completely new materials. In such metallopolymers, it is possible to combine the outstanding properties of both material classes, which brings enormous benefits in some areas of application and can even open the way into new ones [4]. Through the respective choice of metal ion, ligand, and the associated stability of the resulting metal complex, it is possible to control the polymer properties in a targeted manner and to adapt them to the respective requirements [5]. This new class of polymers can be used, for example, in the field of catalysis [6] or opto-electronics [7] and sensors [8]. The implementation of reversible metal-complexes into the polymeric structure makes the synthesis of stimuli-responsive materials easier. Depending on the type of metallopolymer and the utilized metal–ligand interaction, thermally [9], photo-chemically [10], mechanically [11], or redox-reaction [12] induced switching of the materials is possible. In addition to the mentioned areas of possible applications, the utilization of intelligent metallopolymers for the synthesis of polymers with self-healing [13] or shape-memory abilities [14,15] has become more and more popular during the last years. The so-called shape-memory polymers (SMPs) are a very interesting class in the field of intelligent materials. These polymers have the unique ability to “remember” their previously programmed permanent shape. Under certain conditions, it is possible to transform them into a completely new form, the so-called temporary shape, in which they remain until a certain external stimulus act on them. This leads to the recovery of the original determined permanent shape [16]. In order to show shape-memory ability, the polymer must exhibit special structural elements. Firstly, the so-called stable phase is required. The task of this structural unit is, on the one hand, to generate stability. On the other hand, the deformation of this unit generates the driving force for the recovery process. This structural unit can be generated in different ways [17]. The most common strategy is the utilization of covalent [18] or physical [19] crosslinking to generate a permanent polymer network. However, some literature reports also showed that using interpenetrating networks is a possibility to generate the stable phase [20].

The second element is the so-called switching unit, which must have a certain degree of reversibility. Furthermore, it must be possible to influence its reversibility by the desired external trigger to achieve the switching process. Consequently, this part of the polymer network fixes the temporary shape until a trigger is applied [21]. The most common way to generate this unit is the utilization of thermal transitions such as crystallization/melting (*T_m_* based) [22] or the glass transition (*T_g_* based) [23]. Furthermore, reversible bonds, both covalent and supramolecular, can also be used as switching units. In the field of reversible covalent bonds, Diels–Alder [24] or other cycloaddition reactions [25] are prominent examples. For reversible supramolecular interactions, it was possible to utilize hydrogen bonds [26], ionic interactions [27], and metal–ligand complexes [14,15].

The class of shape-memory polymers can further be subdivided. The most common examples, called dual shape-memory polymers, are only able to switch between two shapes. This means that it is only possible to program one temporary shape, which can then be converted back to its original permanent shape by the external stimulus. However, in the last few years, several examples have been presented in literature showing the possibility to program two (or even more) temporary shapes, which can be restored individually [28]. For the programming of several temporary shapes and the corresponding recovery steps, either different stimuli or one stimulus of different intensities could be used. Polymers in which the programming of two temporary shapes is possible are called triple (or, in the case of several temporary shapes, multiple) shape-memory polymers. The design principle for triple shape-memory polymers is almost identical; however, the switching unit needs to be adapted [29]. It was found, for example, that a very broad *T_g_* can be used to program several temporary shapes in a wide temperature range, like Xie et al. presented for Nafion [30]. Furthermore, it is of course also possible to incorporate two or more switching units into the polymer structure and then specifically address them for the fixation and the recovery of the different shapes [23,31]. Exemplarily, Lendlein et al. presented in 2006 the synthesis of triple shape-memory polymers, in which the first switching unit was based on the *T_g_* of the utilized poly(ethylene glycol) chains, while the second one resulted from the crystallization and melting of incorporated poly(caprolactone) domains [23].

In the current literature, there are also several examples for triple or multiple shape-memory polymers bearing a supramolecular structure as one switching unit. For example, Voit et al. synthesized polymers exhibiting triple shape-memory ability based on the *T_g_* of the polymer for the first shape, while the second switching unit was based on hydrogen bonds formed by ureidopyrimidone (upy) side chains [32]. Furthermore, we already presented in previous studies a triple shape-memory polymer, in which one switching unit is based on the glass transition temperature of the polymer network while the second one could be generated utilizing supramolecular crosslinking via triazole-pyridine complexes [18,33]. Other studies, reported the possibility to partially incorporate metal complexes into the polymeric material, enabling the switching of the different shapes utilizing heat for the first and UV-light for the second shape as triggers [34]. Nevertheless, to the best of our knowledge, there are no examples for triple shape-memory polymers in which both switching units are based on two different supramolecular crosslinks.

This study presents the synthesis of supramolecular triple shape-memory polymers, in which the two switching units are formed by two metal complexes with different activation temperatures. The stable phase of the system was achieved by covalent crosslinking of the polymeric network.

## 2. Experimental section

### 2.1. Materials and Methods

All chemicals were used as received from TCI (Eschborn, Germany), Sigma Aldrich (Darmstadt, Germany), Alfa Aesar (Kandel, Germany), Thermo Fisher Scientific (Geel, Belgium), and Acros Organics (Geel, Belgium) if not otherwise stated. All solvents were dried over molecular sieve under nitrogen atmosphere. The stabilizer in the used liquid monomers butyl methacrylate (BMA) and triethyleneglycole dimethacrylate (TEGDMA) was removed over a short aluminum oxide (AlOx) column (neutral AlOx, obtained from Molecula, Darlington, UK). The ligand monomers, 6-(2,2′:6′2″-terpyridin-4′-yloxy)-hexyl methacrylate (**Tpy-MA**) [13] and 11-[4-(pyridine-2-yl)-1*H*-1,2,3-triazol-1-yl]undecanyl-methacrylate (**Triaz-Py-MA**), as well as the model compound, 11-[4-(pyridine-2-yl)-1*H*-1,2,3-triazol-1-yl]undecanyl-acetate (**Triaz-Py**) [35,36], were synthesized according to literature procedures. The reaction schemes are displayed in the Appendix A.

Nuclear magnetic resonance (NMR) spectra were measured using a Bruker AC 250 (250 MHz), Bruker AC 300 (300 MHz), Bruker AC 400 (400 MHz), and a Bruker AC 600 (600 MHz) spectrometers (Billerica, MA, USA) at 298 K if not stated differently. The chemical shift is given in parts per million (ppm on *δ* Scale) related to deuterated solvent. All NMR spectra are presented in the Appendix A.

Elemental analysis (EA) was performed utilizing a Vario El III (Elementar, Langenselbold, Germany). All results of the elemental analyses are summarized in the Appendix A. For the covalently crosslinked polymers and metallopolymer networks, the calculated values are based on the ratio of the monomers utilized for the polymerization.

Isothermal titration calorimetry (ITC) measurements were performed using a standard volume Nano ITC (TA Instruments) at 303 K. The solutions were always prepared prior to use in dry solvents utilizing vacuum dried ligand and metal salt. Blank titrations in dry solvent were performed and subtracted from the corresponding titrations to remove the effect of dilution. The fitting of the measured data was performed with the NanoAnalyze program from TA instruments. The resulting titration plots are displayed in the Appendix A.

Size exclusion chromatography (SEC) measurements were performed on the following setup: Shimadzu with CBM-20A (system controller), DGU-14A (degasser), LC-20AD (pump), SIL-20AHT (auto sampler), CTO-10AC vp (oven), SPD-20A (UV detector), RID-10A (RI detector), PSS SDV guard/1000 Å/1,000,000 Å (5 μm particle size) chloroform/isopropanol/triethyl-amine [94/2/4] with 1 mL/ min at 40 °C, poly(methyl methacrylate) (standard). The resulting SEC curves are displayed in the Appendix A.

Differential scanning calorimetry (DSC) was measured on a Netzsch DSC 204 F1 Phoenix instrument (Selb, Germany) under a nitrogen atmosphere with a heating rate of 20 K min^−1^ (first and second heating cycle) and 10 K min^−1^ (third heating cycle). The resulting DSC curves are displayed in the Appendix A.

Thermogravimetric analysis (TGA) was carried out under normal atmosphere with a heating rate of 10 K min^−1^ using a Netzsch TG 209 F1 Iris (Selb, Germany). The resulting TGA curves are displayed in the Appendix A.

Thermo-mechanical analyses (TMA) were performed on an MCR 301 rheometer from Anton Paar (Graz, Austria) using the convection oven device CTD 450, which covers a temperature range from −150 to 450 °C. Rectangular samples of approximately 30 mm length, 10 mm width, and a thickness ranging between 2 and 5 mm were measured using a solid rectangular fixture (SRF) designed for solid samples. After fixing the samples, a free length ranging between approximately 15.0 and 20.5 mm resulted in the experiments. All TMA measurements (dual and triple) were performed analogically to literature procedure [33,37]. All resulting TMA plots are displayed in the Appendix A.

FT-Raman spectroscopy measurements were performed on a Multispec Fourier-transform Raman-Spectrometer (Bruker Corporation, Billerica, Massachusetts, United States of America) in the range between 100 and 4000 cm^−1^ with a spectral resolution of 4 cm^−1^. The Raman excitation light at 1064 nm was provided by a Nd:YAG laser (Klastech DeniCAFC-LC-3/40, Dortmund, Germany). The laser power at the sample was set to 100 mW and 512 accumulated scans were recorded for each sample to improve the signal-to-noise ratio. Full spectra of all samples, including reference materials are shown in the Appendix A.

### 2.2. Synthesis of the Polymer Network P1 via Free Radical Polymerization

The polymerization was carried out in a 50 mL one-neck-round bottom flask. 2,2′-Azo*bis*(2-methylpropionitrile) (AIBN) (57.7 mg, 0.35 mmol) as initiator, BMA (4.0 g, 28.13 mmol), TEGDMA (0.4 g, 1.41 mmol), 6-(2,2′:6′2″-terpyridin-4′-yloxy)-hexyl methacrylate (**Tpy-MA**) (1.17 g, 2.81 mmol), and 11- [4-(pyridine-2-yl)-1*H*-1,2,3-triazol-1-yl]undecanyl-methacrylate (**Triaz-Py-MA**) (1.1 g, 2.81 mmol) were added into the flask. Afterwards, the required amount of dimethylformamide (DMF) (18 mL) was added to reach the desired concentration of 2 mol L^–1^. The monomer to initiator ratio was 100 to 1. The reaction mixture was purged with nitrogen for 30 min and afterwards stirred overnight in a preheated oil bath at 70 °C. The resulting swollen polymer network was washed with water, acetone, and chloroform to remove the DMF and remaining unpolymerized monomers. The obtained polymer was dried for 24 h under vacuo at a temperature of 40 °C.

**P1**: Elemental analysis: calcd. C 68.16, H 8.96, N 5.93; found. C 62.40, H 7.91, N 5.87.

### 2.3. Synthesis of the Metallopolymer Networks

To synthesize the metallopolymer networks **P1-Fe/Fe**, **P1-Co/Co**, **P1-Fe/Zn**, and **P1-Co/Zn**, the covalently crosslinked polymer network **P1** was placed in a 20 mL vial and was swollen in dichloromethane (5 mL). The calculated amount of the metal salt was separately dissolved in methanol (2 mL) and added to the swollen polymer network. In the case of the utilization of two different salts for the synthesis of **P1-Fe/Zn** and **P1-Co/Zn**, first the salt for the formation of the strong terpyridine complexes and afterwards the zinc triflate for the formation of the labile triazole-pyridine complexes was added. The resulting metallopolymer networks were dried for 24 h under vacuo at a temperature of 40 °C to remove the solvents. The utilized quantities for the synthesis are listed in Table 1.

**P1-Fe/Fe**: Elemental analysis: calcd. C 61.11, H 8.69, N 5.32, S: 0.89; found. C 56.74, H 7.27, N 5.12, S: 1.35

**P1-Co/Co**: Elemental analysis: calcd. C 65.30, H 8.58, N 5.52; found. C 58.61, H 7.47, N 5.00

**P1-Fe/Zn**: Elemental analysis: calcd. C 60.71, H 8.25, N 5.24, S: 1.32; found. C 57.29, H 7.18, N 5.64, S: 1.44

**P1-Co/Zn**: Elemental analysis: calcd. C 62.73, H 8.19, N 5.34, S: 0.90; found. C 57.06, H 7.20, N 5.34, S: 1.03.

## 3. Results and Discussion

### 3.1. Isothermal Titration Colorimetry

For the utilization of two different metal-complexes as two switching units in a triple shape-memory polymer, the complex association constant *K_α_* of the formed metal complexes plays an important role. In order to determine those values, as well as the stoichiometry of the formed complexes, isothermal titration calorimetry (ITC) measurements were performed with different model systems and metal salts. Based on former studies [14,18,37], it was decided to utilize 2,2′:6′,2″-terpyridine (**Tpy**), known for the formation of stable complexes with higher activation temperatures with several metal salts. For the second switching unit, which should feature a lower activation temperature, a pyridine-triazole ligand (**Triaz-Py)** was applied. The synthesis of the **Triaz-Py** ligand without any polymerizable group, which was utilized as model system in the ITC investigations, was performed according to a literature procedure [35,36]. The reaction scheme is presented in the Appendix A). The results of the ITC measurements with the two different ligands and several salts are presented in Table 2 and Appendix A. For FeSO_4_ and Co(OAc)_2_, these results revealed the formation of stable complexes with terpyridine, while the complexes with triazole-pyridine are rather weak. Additionally, it was found that Zn(TFMS)_2_ also forms labile complexes with the **Triaz-Py**, which should feature a rather low activation temperature as well. Regarding the triazole-pyridine ligand, the association constant determined for the formation of the complex with Zn(TFMS)_2_ was the lowest. The highest value in this context was found with Co(OAc)_2_, while the determined value for FeSO_4_ was between those two. From this, the following complex stability should be derived based on the metal salt utilized: Zn < Fe < Co. In contrast, the terpyridine complex formed with iron sulfate has a significantly higher value for K_α_ compared to the cobalt *bis*-complex and should, therefore, also be significantly more stable.

### 3.2. Synthesis of a Polymer Network and Metallopolymer Networks

For the synthesis of the covalent crosslinked polymer network (**P1**) bearing two different ligands in the side chains, the required ligand monomers, 6-(2,2′:6′2″-terpyridin-4′-yloxy)-hexyl methacrylate (**Tpy-MA**) and 11-[4-(pyridine-2-yl)-1*H*-1,2,3-triazol-1-yl]undecanyl-methacrylate (**Triaz-Py-MA**), were synthesized according to literature procedures [13,35,36] (see Appendix A). Subsequently, the two ligand monomers (10% of each) were polymerized via free radical polymerization together with *n*-butyl methacrylate (BMA) as main monomer and 5% triethyleneglycol dimethacrylate (TEGDMA) as covalent crosslinker using 2,2′-azo*bis*(2-methylpropionitrile) (AIBN) as initiator. The schematic representation of the synthesis of **P1** is displayed in Figure 1a. The obtained polymer revealed swelling; however, it was insoluble in any common solvent (e.g., chloroform, DMF, tetrahydrofuran, or acetone) indicating the successful synthesis of a covalent crosslinked polymer network. The elemental analyses revealed a nitrogen content of 5.87%, which fits quite well with the calculated value of 5.93%. The latter value was calculated based on the utilized monomer ratios during the polymerization. For this reason, it can be assumed that the polymer network has the targeted composition. An exact determination of the composition was not possible due to the above-mentioned insolubility.

For the formation of metallopolymer networks different salt combinations, based on the ITC results, were added to **P1**, which was swollen in chloroform. The stoichiometries determined during the ITC measurements were rounded, since a stoichiometry of 1.6 was determined for the formation of the triazole-pyridine complex with iron(II) sulfate or cobalt(II) acetate. However, literature reports also indicated the formation of 2:1 complexes [38]. For this reason, a ratio of 2 to 1 was utilized as the ratio of ligand (either triazole-pyridine or terpyridine) to metal salt. The schematic representation of the synthesis of the metallopolymer networks is presented in Figure 1b. Iron(II) sulfate or cobalt(II) acetate as salt were used for the formation of the stronger complexes with the terpyridine ligands. Furthermore, those two salts and, additionally, zinc(II) trifluoromethane sulfonate were utilized for the formation of the weaker complexes formed by the triazole-pyridine ligands. The required amount of salt(s) was calculated based on the ligand content of **P1**, which was assumed to be the same as in the monomer mixture indicated by the nitrogen content found in elemental analyses. In total, four different covalently and supramolecular crosslinked metallopolymer networks **P1-Fe/Fe**, **P1-Co/Co**, **P1-Fe/Zn,** and **P1-Co/Zn** were prepared.

### 3.3. Raman Spectroscopic Investigations

To confirm the formation of the metal complexes inside the polymers and to investigate the binding environment, FT-Raman spectroscopy was performed on the polymer **P1** and its derivatives. The Raman spectra of **P1** and the metallopolymer networks **P1-Fe/Fe**, **P1-Co/Co**, **P1-Fe/Zn,** and **P1-Co/Zn** are depicted in Figure 1 (full spectra are presented in the Appendix A). Raman spectroscopy is a well-established tool to investigate active centers inside polymer environments and was already utilized in previous work to investigate terpyridine and trityl-histidine complexes of zinc, iron, and nickel ions [14,37]. Therefore, changes to the Raman spectrum of **Tpy** are already well known. Namely, the aromatic vibrations at 996 cm^−1^, 1320 cm^−1^, and 1566 cm^−1^ blue shift due to a conformational change of the **Tpy** ligand upon metal binding, causing a trans/cis isomerization [14]. To investigate the changes in the Raman spectra upon binding to the **Triaz-Py** inside the polymer environment of **P1**, model polymers of BMA functionalized with only one of the two ligands were investigated (see Appendix A). The synthesis of those model polymers and metallopolymer networks is presented in the Appendix A. The observed changes for **Triaz-Py** unfortunately lie in regions similar to those relevant for **Tpy**. This is due to the similar chemical nature of both moieties (both are *N*-heteroaromatic structures), leading to overlapping bands. Nevertheless, marker bands for **Triaz-Py** are a characteristic blueshift of the vibrations at 980, 1550, and 1600 cm^−1^ upon addition of a metal salt. Interestingly, these shifts, particularly for the bands at 1550 and 1600 cm^−1^ are much stronger for zinc compared to cobalt and iron. Differences in coordinating behavior of zinc and iron to **Triaz-Py**-like ligands are known from literature and are attributed to the longer bond lengths and geometric flexibility of the d^10^ Zn^2+^ ion compared to the d^6^ Fe^2+^ [39]. Since iron and cobalt generally behave chemically similar, it is reasonable to assume that this difference is also applicable to zinc/cobalt and therefore explains the different strength of the band shifts for these ions. In the context of this work, these differences prove very helpful, since they should allow the differentiation of **Zn-Triaz-Py** from **Co/Fe-Triaz-Py**, thereby enabling ion sensitive spectroscopy in the mixed ion derivatives of **P1**.

Formation of the **Tpy**-metal (**M-Tpy**) complexes is easily detectable for all metallopolymer networks due to the strong marker band situated at around 1030 cm^−1^, that is selective to **M-Tpy** moieties. For the formation of **Triaz-Py** metal complexes (**M-Triaz-Py**), the confirmation of the selection is less trivial due to the highly overlapping features in the region around 1000 cm^−1^. The marker band at ca. 1000 cm^−1^ for **M-Triaz-Py** coincides with the band for unbound **Tpy**, which obfuscates its presence. Nevertheless, the presence of a band at 1000 cm^−1^ after addition of the metal salts is a clear indicator that **Triaz-Py** is bound to a metal. For the model polymers **P2** (containing only **Tpy**) as well as in previous studies [37], it was found that the band completely vanishes upon formation of the **M-Tpy** complexes. Therefore, the remaining intensity at this position can be attributed to the formation of **M-Triaz-Py**. This confirms the successful formation of a **Tpy** as well as a **Triaz-Py** metal complexes in all metallopolymer networks of **P1**.

Furthermore, for the mixed networks **P1-Fe/Zn** and **P1-Co/Zn**, the region between 1500 and 1650 cm^−1^ (Figure 1, right panel) is also of interest. As mentioned above, the changes to the Raman signature of **Triaz-Py** are much more pronounced for zinc compared to iron and cobalt here. Indeed, an additional band at ca. 1570 cm^−1^ can be seen for the metallopolymer networks, indicating successful formation of specifically the **Zn-Triaz-Py** complexes inside the polymers. This proves the selective complexation of both metal salts to the respective moieties, necessary for the triple shape-memory behavior.

### 3.4. Investigation of the Thermal Properties

To investigate the thermal properties, which represent key parameters for thermal triggered shape-memory process, differential scanning calorimetry (DSC) measurements and thermogravimetric analysis (TGA) were carried out with the synthesized polymer and metallopolymer networks. The results are summarized and displayed in Table 3 as well as Figure 2 (further information in the Appendix A).

Within the TGA investigation, it is possible to determine the degradation temperature (*T_d_*). This value is important to find out up to what temperature the polymer can be switched without being decomposed. All covalently crosslinked metallopolymer networks **P1-Fe/Fe**, **P1-Co/Co**, **P1-Fe/Zn**, and **P1-Co/Zn** revealed degradation temperatures above 200 °C enabling the switching based on the more stable terpyridine complex at high temperatures.

Furthermore, within the DSC investigation, it was possible to determine the glass transition temperature (*T_g_*) of all samples. Above this temperature, the material becomes soft and flexible enough to be deformed. It turned out that the metallopolymer networks synthesized do not reveal a sharp *T_g_*, but rather show a broad area in which they soften (see Table 3) unlike what was found for the polymer **P1**, which presumably goes in hand with the activation of the metal complexes (see Appendix A). The measurements revealed the lowest range of softening for the sample **P1-Fe/Fe** while the highest range was determined for the sample **P1-Fe/Zn**. The samples **P1-Co/Co** and **P1-Co/Zn** revealed relatively comparable softening ranges in the DSC investigations, even though the turnover point for the sample **P1-Co/Co** was, at 114 °C, slightly higher compared to the value of 101 °C, determined for sample **P1-Co/Zn**. Nevertheless, all four samples showed a softening in comparable ranges.

### 3.5. Investigation of the Shape-Memory Abilities

In order to investigate the shape-memory abilities of the synthesized covalently crosslinked metallopolymer networks, a permanent shape had to be determined. This was performed analogous to a literature procedure [14]. For this reason, the samples **P1-Fe/Fe**, **P1-Co/Co**, **P1-Fe/Zn,** and **P1-Co/Zn** were filled into a self-manufactured mold and hot pressed at a temperature of about 150 to 160 °C and a weight of about 1 to 4 t. In this way, a rectangular specimen could be obtained for each sample (exemplarily, see Figure 3a). All synthesized covalently crosslinked metallopolymer networks showed triple shape-memory behavior during a manually performed test. A photo series of the sample **P1-Co/Zn** is exemplarily presented in Figure 3. For the remaining three samples, it is displayed in the Appendix A. In the beginning of this test, the sample was heated to 140 °C, a temperature above the activation temperature of both complexes. At this temperature, the sample was elongated and cooled below the activation temperature of the terpyridine complex to fix the first temporary shape (see Figure 3b) via the supramolecular crosslinking of the deactivation of terpyridine complexes. Furthermore, the sample was twisted at a temperature of 80 °C, at which only the triazole-pyridine complexes should be activated, and afterwards cooled to room temperature to determine the second temporary shape (see Figure 3c), fixed by the reformed and non-activated triazole-pyridine complexes. The metallopolymer network stayed in this elongated and twisted shape until it was again heated to 80 °C, leading to the recovery of the temporary shape A (see Figure 3d), by activating the weaker supramolecular crosslinking. Heating the polymer afterwards again to the initial temperature of 140 °C, at which the terpyridine complexes are activated again, induced the restoration of the original permanent shape under shrinking of the metallopolymer network (see Figure 3e).

Furthermore, to quantify the shape-memory ability and to compare the different metallopolymer networks with each other, thermo-mechanical analyses (TMA) were performed. Within those measurements it is possible to calculate the strain fixity (*R_f_*) and strain recovery rates (*R_r_*) according to Equations (1) and (2). During this measurement, the sample in its permanent shape (*γ_A_*) was deformed (*γ_B,load_*) above the switching temperature (*T_sw_*) followed by a cooling step below *T_sw_*. Afterwards, the stress was completely removed from the sample leading to the fixed temporary shape (*γ_B_*). Heating the sample again to the initial temperature leads to the recovered permanent shape (*γ_A,rec_*). While the strain recovery rate quantifies the ability of the material to fix a mechanical deformation, the strain recovery rate gives information about the quality of the restorage of the original shape.
*R_f_* = γ_B_/γ_B,load_ × 100%(1)
*R_r_* = (γ_B, load_ − γ_A, rec_)/(γ_B,load_ − γ_A_) × 100%(2)

In the beginning, for each metallopolymer network, standard TMA measurements at two or three switching temperatures were performed to investigate the dual shape-memory abilities of the materials at those temperatures. Exemplarily, the two resulting TMA-plots of the sample **P1-Fe/Fe** are displayed in Figure 4a. All results of this investigation are presented in the Appendix A. Furthermore, the calculated strain fixity and recovery rates are reported in Table 4. Additionally, the triple shape-memory abilities were also investigated via TMA measurements. Exemplarily, the resulting TMA plot is displayed in Figure 4b for the sample **P1-Fe/Fe**. The remaining plots are presented in the Appendix A. In the beginning, the sample in its permanent shape (*γ_A_*) was heated to the first switching temperature (*T_sw1_* = 150 °C) which was above the activation temperature of the more stable terpyridine complexes. At this temperature, the sample was twisted to determine the first temporary shape (*γ_B,load_*) followed by a cooling step to the second switching temperature (*T_sw2_* = 110 °C or 100 °C). Afterwards, the shear stress was completely removed from the sample followed by an annealing step to determine the fixed first temporary shape (*γ_B_*). The sample was then deformed again, now at the lower temperature of 110 °C, respectively 100 °C, to program the second temporary shape (*γ_C,load_*). Cooling the sample afterwards to 30 °C, releasing the shear stress from the sample and annealing it, leads to the second fixed temporary shape (*γ_C_*). To investigate the regeneration of the different shapes, the sample was heated again under stress free conditions, to firstly 110 °C or 100 °C inducing the recovery of the first temporary shape (*γ_B,rec_*), and afterwards to the initial temperature of 150 °C leading to the regeneration of the original permanent shape (*γ_A,rec_*). With Equations (3)–(8), it is possible to calculate the strain fixity and recovery rates for the single steps as well as the overall values which are summarized in Table 4.


*R_f1_ (A* → *B)* = (γ_B_ − γ_A_)/(γ_B, load_ − γ_A_) × 100%(3)
*R_f1_ (B* → *C)* = (γ_C_ − γ_B_)/(γ_C, load_ − γ_B_) × 100%(4)
*R_f, total_ (A* → *C)* = (γ_C_ − γ_A_)/(γ_C, load_ − γ_A_) × 100%(5)
*R_r1_ (C* → *B)* = (γ_C_ − γ_B, rec_)/(γ_C_ − γ_B_) × 100%(6)
*R_r2_ (B* → *A)* = (γ_B, rec_ − γ_A, rec_)/(γ_B_ − γ_A_) × 100%(7)
*R_r, total_ (C* → *A)* = (γ_C_ − γ_A, rec_)/(γ_C_ − γ_A_) × 100%(8)


The measurements revealed a moderate strain fixity rate (70 to 87%) for the fixation of the first temporary shape at 110 °C. This is possibly due to the fact that the terpyridine complexes are not yet completely deactivated at this temperature and that the determined form is therefore not completely fixed. The samples containing two different salts (**P1-Fe/Zn** and **P1-Fe/Co**) revealed herein higher fixity rates compared to **P1-Fe/Fe** and **P1-Co/Co**, which contain only one metal salt. Presumably, the utilization of two different metal salts for the synthesis results in a better deactivation of the terpyridine complex at 110 °C. The current working hypothesis is that for the samples containing the same salt in both complexes, either mixed complexes might be formed as intermediates at 110 °C during the exchange reaction of both metal complexes or different superstructures and morphologies are present due to the different metal ions utilized. The kind of the switching unit seems to influence the reversible behavior of the stable phase (although this is the same for the compared polymers). A similar behavior could already be shown for a different ligand system [37].

Nevertheless, the calculated values of *R_f1_* are quite good for triple shape-memory polymers. In order to investigate whether a reduction in temperature could counteract this phenomenon and, therefore, increase the fixity rate (*R_f1_*), an additional measurement was carried out with the samples **P1-Fe/Fe** and **P1-Co/Co**, in which *T_sw2_* was reduced from 110 to 100 °C. As expected, the fixity rate of this step could be significantly increased. The fixation of the second temporary shape was excellent for all samples within a calculated strain fixity rate for this step of 100%. Furthermore, the calculated recovery rates of the samples showed satisfying results. For the measurement performed with *T_sw2_* = 110 °C, the best performing sample regarding the recovery rates was **P1-Fe/Fe**, which revealed *R_r_* values of about 95% for both recovery steps. Moreover, the sample **P1-Co/Co** revealed good recovery behavior, in particular during the first recovery step. This is exactly the opposite behavior to the first fixation step. The fact that these two samples fixed worse in the first step seems to in turn improve the recovery behavior, which presumably results from the behavior described above. Matching this, the reduction of *T_sw2_* to 100 °C also reduced the calculated value for *R_r1_*. It has to be noted that during the second recovery step for the sample **P1-Co/Zn** and **P1-Co/Co** (at *T_sw2_* = 100 °C), a recovery rate above 100% (*R_f2_* = 109.4%, respectively 104.2%) was found. This can be explained by an incomplete recovery during the first recovery step, which is often noticed during the characterization of triple shape-memory polymers [23,33,40]. The sample **P1-Fe/Zn** in general revealed, compared to the other three samples, the lowest recovery rates which may have resulted from its thermal properties. Looking at the previous DSC investigations, it becomes clear that this sample has the highest *T_g_*. Therefore, this sample can be considered as less soft as the other three polymer networks resulting in a lower recovery. Nevertheless, these lower recovery rates above 83% are still within a good range if compared to values from the literature for other triple shape-memory polymer systems.

In literature, there are several examples for triple and even multiple shape-memory polymers [23,32,40,41]. However, a system that is very similar in structure to the herein presented triple shape-memory metallopolymer networks, in which both temporary forms can be switched via two different reversible supramolecular metal–ligand interactions, is to the best of our knowledge not available. Comparable reports describe polymers featuring supramolecular crosslinks via hydrogen bonds. These materials exhibit good triple shape-memory abilities [32]. Another example is the utilization of side chain liquid crystallization for the creation of triple SMPs. The TMA investigation of this kind of material revealed comparable results, with *R_f1_* values from 56 to 79%, *R_f2_* values near 100% and recovery rates similar to those of the presented metallopolymer networks [40]. Overall, the synthesized covalently crosslinked metallopolymer networks **P1-Fe/Fe**, **P1-Co/Co**, **P1-Fe/Zn,** and **P1-Co/Zn** revealed comparable shape-memory properties to other triple shape-memory polymers known from literature.

## 4. Conclusions

In the scope of this study, we were able to successfully synthesize triple shape-memory metallopolymer networks in which the switching of the two temporary shapes could be realized by activating and deactivating metal–ligand interactions of different stability. The synthesized materials were analyzed in a detailed fashion. Thermogravimetric analyses and differential scanning calorimetry were utilized to investigate the thermal properties. Additionally, within isothermal titration calorimetry and FT-Raman spectroscopy, it was possible to investigate the different metal–ligand interactions utilized for the reversible crosslinking. The performed thermo-mechanical analyses revealed very good dual and triple shape-memory abilities for all synthesized metallopolymer networks. Results showed that the sample **P1-Fe/Fe** is the best performer during the recovery step (both recovery rates above 95%). In contrast, the samples containing two different salts **P1-Fe/Zn** and **P1-Co/Zn** revealed the highest strain fixity rates for the first fixation step. In summary, the presented metallopolymer networks exhibit a good triple shape-memory performance.

## Data Availability

The original data can be obtained upon request from the corresponding author.

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
