# Peer review of "Synthesis and Characterization of Metallopolymer Networks Featuring Triple Shape-Memory Ability Based on Different Reversible Metal Complexes"

_polymers, 2022, doi:10.3390/polym14091833_

Round 1

Reviewer 1 Report

Meurer et al. wish to present the synthesis of novel metallopolymer networks with shape-memory ability. For that, they used covalently crosslinked polymers with two additional ligands, namely terpy-type (tridentate) ligand and pyridine-triazole type (bidentate) ligand. They subsequently added various metal salts (Fe, Zn, Co) which lead to selective formation of complexes, with two different association constants. The constants were evaluated by means of isothermal titration calorimetry. These two crosslinks feature different activation temperatures and therefore can act as two individual switching units. Overall, this enables the fixation and recovery of material shapes.

Overall, this is a nice contribution that deserves to be published in a journal such as Polymers. The work was done properly and the supporting information is complete. I recommend acceptance of the paper.

Reviewer 2 Report

This is an interesting work by Schubert co-workers on the triple shape-memory ability of metallopolymers. The authors described the synthesis and characterization of four different covalently, and supramolecular crosslinked metallopolymer networks bearing pyridine-triazole ligand was applied (Triaz-Py). They demonstrated that two temporary shapes could be switched via activating and deactivating metal-ligand interactions. Overall, I support the publication of this paper in polymers after the following minor changes:

  1. Not all the acronyms used in this paper have been defined the first time that they were used. I suggest checking this in the manuscript.
  2. Page 5, line 236: “The obtained polymer revealed swelling; however, it was insoluble in any common solvent indicating the 237 successful synthesis of a covalent crosslinked polymer network” Which solvents ( polar/non-polar) and in what quantity?
  3. In SI, replace the word Tiaz by Triaz.
  4. Scheme 1 and S3: What does the blue dotted line denote?
  5. There is no discussion on SEC of the polymers.

Reviewer 3 Report

Manuscript Number: polymers-1686489

Title: Synthesis and characterization of metallopolymer networks featuring triple shape-memory ability based on different reversible metal complexes

Author(s): Josefine Meurer , Thomas Bätz , Julian Hniopek , Milena Jäger , Stefan Zechel , Michael Schmitt , Juergen Popp , Martin D Hager , Ulrich S. Schubert *

-----------------------------------------------------------------------

General Comments:

This manuscript reports a study about a metallopolymer network with a triple shape-memory ability. Specifically, the covalently crosslinked polymers with two different ligands is constructed by radical polymerization. Detailed chemical structure, microstructure and mechanical properties of series foams were evaluated via 1H NMR, FTIR, TGA, TMA, etc. A major conclusion was the shape-memory abilities of the networks. Overall, this report presents a comprehensive work though some details might be questionable. This has somewhat deteriorated the technical merit of this work. Some specific comments are provided below for the authors’ consideration to improve this manuscript before its publication.

  1. The 1H NMR evidenced the structure, but it would provide more solid proof of well-controlled content by giving a peak assignment and integration of selected peaks to identify the content of Tpy-MA: Triaz-Py-MA for later discussion.
  2. It would be straightforward to provide the molar content of metal salt than mass, and it is interesting to know the content of ligands in networks.
  3. Rather than listing the Td Tg of the control group P1 in a table, it is better to present the TGA and DSC curve.
  4. As the P1-Fe/Fe outperformed other groups, it would be more convincing to further discuss the mechanisms by associating the structure or previous work.
  5. It is interesting to know the crosslink density of the metallopolymer network and the change in mechanical properties versus the control group P1.
